# CD34+ Cell Dose, Measurable Residual Disease, and Outcome After Myeloablative HLA-Matched Peripheral Blood Hematopoietic Cell Transplantation for Adults with Acute Myeloid Leukemia

**DOI:** 10.3390/cancers17142323

**Published:** 2025-07-12

**Authors:** Margery Gang, Megan Othus, Anne-Chloe Olix, Kate A. Markey, Derek L. Stirewalt, Laura S. Connelly-Smith, Stephanie J. Lee, Filippo Milano, Roland B. Walter

**Affiliations:** 1Hematology/Oncology Fellowship Program, Fred Hutchinson Cancer Center, University of Washington, Seattle, WA 98109, USA; 2Public Health Science Division, Fred Hutchinson Cancer Center, Seattle, WA 98109, USA; mothus@fredhutch.org (M.O.); lsconnel@fredhutch.org (L.S.C.-S.); sjlee@fredhutch.org (S.J.L.); 3Translational Science and Therapeutics Division, Fred Hutchinson Cancer Center, Seattle, WA 98109, USA; aolix@fredhutch.org (A.-C.O.); kmarkey@fredhutch.org (K.A.M.); dstirewa@fredhutch.org (D.L.S.); fmilano@fredhutch.org (F.M.); rwalter@fredhutch.org (R.B.W.); 4Department of Medicine, Division of Hematology and Oncology, University of Washington, Seattle, WA 98109, USA; 5Department of Laboratory Medicine and Pathology, University of Washington, Seattle, WA 98109, USA

**Keywords:** acute myeloid leukemia, allogeneic hematopoietic cell transplantation, donor cell dose, CD34-positive cells, measurable residual disease, relapse

## Abstract

In acute myeloid leukemia, a hematopoietic cell transplant can offer cure. However, relapse remains the most common cause of treatment failure, especially in patients with measurable residual disease (MRD) at time of transplant. It remains unclear how the number or types of immune cells in the donor graft impacts transplant outcomes and relapse risk and whether these effects vary based on pre-transplant MRD status. In our study, we evaluated two immune cell types, CD34+ cells and CD3+ T cells, and their impact on outcomes after transplantation with myeloablative conditioning. We found that receiving higher doses of CD34+ cells was associated with lower relapse risks and improved survival, but there appeared to be a threshold effect. CD3+ cell doses did not impact relapse or survival but were linked to higher risk of chronic graft-versus-host disease. These findings may help guide decisions about donor cell selection to improve transplant outcomes for AML.

## 1. Introduction

With continued advances in supportive care, relapse has become the dominant cause of treatment failure for adults with acute myeloid leukemia (AML) undergoing allogeneic hematopoietic transplantation (HCT) [1,2,3,4,5]. Still, the likelihood of leukemia recurrence varies widely across patients. Among the many recognized risk factors, measurable residual disease (MRD) at the time of HCT has been consistently linked to increased risk of relapse and worse survival [6,7]. Some studies have also suggested that the cellular composition of the donor graft may impact post-HCT outcomes in patients with hematologic malignancies [8,9,10,11,12,13,14,15,16,17,18,19,20]. Most notably, higher graft CD34+ cell doses have been associated with a decreased relapse incidence in many studies, with some also showing improved relapse-free survival (RFS) and/or overall survival (OS) [8,9,10,17,19,20]. Among these, a multicenter prospective study of 932 recipients of unrelated donor peripheral blood stem cells (PBSCs) found higher doses of CD34+ cells resulted in rapid engraftment and improved OS without increased risk of graft-versus-host disease (GVHD) [21]. However, others yielded contradictory conclusions with either unaffected or worsened post-HCT outcomes with higher CD34+ cell composition in the graft [22,23,24,25,26], leading some centers to cap infused cell dose. Included in the latter is a retrospective European Society for Blood and Marrow Transplantation (EBMT) registry analysis, in which no correlation between CD34+ cell dose and post-HCT relapse or OS was found in adults receiving reduced intensity conditioning (RIC) allogeneic HCT for AML [27]. The impact of CD3+ T cells on post-HCT outcomes also remains uncertain. In some reports, high CD3+ T cell doses have been linked to increased incidence of acute GVHD [27,28], whereas others noted improved OS without increased risk of severe GVHD [29,30]. However, differences in the clinical situations, specific dose cutoffs, and included diseases complicate comparisons and data interpretation. Moreover, it is unknown whether there is an interplay between the presence of pre-HCT MRD, donor cell doses, and relapse risk. To address this uncertainty, we investigated the relative impact of CD34+ and CD3+ cell doses on outcomes of myeloablative conditioning (MAC) 10/10 HLA-matched PBSC HCT in patients with myelodysplastic neoplasm (MDS)/AML or AML in first morphologic remission with and without detectable MRD in pre-HCT bone marrow specimens.

## 2. Materials and Methods

### 2.1. Patients

For study inclusion, we utilized an electronic database to identify all adults ≥ 18 years of age with MDS/AML or AML (as defined by 2022 International Consensus Classification [ICC]) who underwent MAC and received either 10/10 HLA-matched sibling or unrelated donor mobilized PBSC allografts while in first morphologic remission (<5% blasts in bone marrow) between April 2006 and March 2023 at the University of Washington/Fred Hutchinson Cancer Center (FHCC) [31]. A total of 384 patients were identified who agreed to their data being used for research purposes. Data/follow-up was current as of 1 May 2025. This retrospective analysis was approved by the FHCC Institutional Review Board (#2562).

### 2.2. Data Collection

The 2022 European LeukemiaNet (ELN) criteria were used to classify cytogenetic disease risk at diagnosis, define secondary AML, and categorize blood counts before HCT (i.e., CR, CRh/CRi, MLFS) [32]. The HCT-specific comorbidity index (HCT-CI) was calculated as described [33]. The treatment-related mortality (TRM) score (online calculator: https://trmcalculator.fredhutch.org) was computed with clinical and laboratory data from the first day of conditioning therapy as carried out before [34,35]. Post-HCT maintenance therapy was not routinely used other than in a subset of patients with FLT3- or IDH-mutated neoplasm after appropriate small molecule inhibitors became available. Treatment responses were categorized using 2022 ELN criteria, except that relapse was defined as emergence of >5% blasts by morphology or flow cytometry in blood or bone marrow, re-emergence of cytogenetic abnormalities seen previously, or presence/emergence of any level of disease if leading to a therapeutic intervention [32,36]. MRD testing by 10-color multiparameter flow cytometry (MFC MRD) was performed routinely on bone marrow specimens obtained during the pre-HCT workup. The methodology of the MFC MRD assay has remained essentially unchanged throughout the study period, with any level of MRD considered positive, consistent with prior analyses and the performance characteristics of the assay [36,37,38,39,40]. Notably, the presence of molecular alterations by next-generation sequencing (NGS) assays alone (i.e., without evidence of MFC MRD) was not used to determine MRD status.

### 2.3. Graft Composition

Aliquots of the graft samples were evaluated in the laboratory prior to cryopreservation. PBSC graft doses were calculated based on patients’ actual body weight at the time of HCT. MFC was used to enumerate CD34+ cells, lymphocyte subsets (CD3+ T cells, CD4+ T cells, CD8+ T cells, regulatory T cells [Tregs], CD20+ B cells, and natural killer [NK] cells), and CD14+ monocytes. In situations where cell doses were quantified by multiple flow cytometry panels, an averaged cell dose was computed and used for analysis. Per institutional policy, apart from small aliquots retained for possible donor lymphocyte infusions in unrelated donors, all collected hematopoietic cells were infused.

### 2.4. Conditioning Regimens, GVHD Prophylaxis, and Supportive Care

Conditioning regimens were selected in accordance with institutional protocols. MAC/non-MAC regimens were defined as described [35]. GVHD prophylaxis was categorized as either calcineurin inhibitor (CNI; tacrolimus or cyclosporine) and methotrexate (MTX) based, CNI with mycophenolate mofetil (MMF) based, post-transplantation cyclophosphamide (PTCy), or other. The overall burden and grading of acute GVHD was measured with average acute GVHD Activity Index scores [41]. Chronic GVHD was diagnosed and graded using National Institutes of Health (NIH) consensus criteria [42]. Information on post-transplant outcomes was captured via Long-Term Follow-Up Program through medical records from our outpatient clinic and local clinics that provided primary care for patients.

### 2.5. Statistical Methods

Unadjusted probabilities of RFS (events  =  relapse and death) and OS (event  =  death) were estimated using the Kaplan–Meier method. Probabilities of relapse, non-relapse mortality (NRM), and acute and chronic GVHD were summarized using non-parametric cumulative incidence estimates. NRM was defined as death without prior relapse and was considered a competing risk for relapse, while relapse was a competing risk for NRM. Relapse and non-relapse deaths were competing risks for acute and chronic GVHD. Categorical patient characteristics were compared using Fisher’s exact test and quantitative characteristics were compared with the Wilcoxon rank sum test. Associations with RFS and OS were assessed using Cox regression models; cause-specific regression models were used for relapse, NRM, and acute and chronic GVHD. We excluded all patients with missing CD34+ (*n* = 26) or CD3+ (*n* = 26) cell dose data for respective analyses. Potential non-linear associations between CD34+ and CD3+ cell doses and endpoints (relapse, relapse-free survival, overall survival, and non-relapse mortality) were modeled with a log-transformed continuous variable with restricted cubic spline curves within a Cox regression model. CD34+ and CD3+ cell doses were treated as continuous variables and CD34+ cell doses were transformed into a dichotomous variable after calculating the optimal cutoff value by maximally selected rank statistic. The optimal equal-HR method was used to determine two optimal cutpoints of U-shaped relationships with survival outcomes [43]. Interactions between CD34+ cell dose and post-HCT outcomes were also evaluated through stratified analysis by MRD status and a model with an interaction term to assess statistical significance. Two-sided *p*-values are reported. Statistical analyses were performed using R version 4.4.2 (http://www.r-project.org).

## 3. Results

### 3.1. Study Cohort and Transplant Characteristics

In our analyses, we included 384 adults with AML (*n* = 316, 82%) or MDS/AML (*n* = 68, 18%) who underwent MAC 10/10 HLA-matched sibling or unrelated donor mobilized PBSC HCT in first morphologic remission between 4/2006 and 3/2023 at our institution and agreed to their data being used for research purposes. Their patient, transplant, and graft characteristics, stratified by pre-HCT MFC MRD status, are summarized in Table 1. As anticipated for candidates selected for MAC HCT, patients were generally medically fit, as assessed by the HCT-CI and TRM scores. Most patients (*n* = 272, 71%) received CNI plus methotrexate-based regimens as GHVD prophylaxis, with the remainder receiving CNI plus MMF-based regimens (*n* = 64, 17%), or PTCy (*n* = 37, 10%). Post-HCT maintenance therapy (hypomethylating agent and/or FLT3 inhibitor or IDH inhibitor) was used in 31 (8%) of patients.

Among the 384 patients, 76 (20%) had pre-HCT MFC MRD (i.e., were “MRD^pos^”). These patients differed from the 308 (80%) MRD^neg^ patients in that they were older (*p* = 0.005), had higher TRM scores (*p* < 0.001), and more likely had adverse risk disease (50.0% vs. 25.0%, *p* < 0.001), MDS/AML rather than AML (34.2% vs. 13.6%, *p* < 0.001), secondary disease (35.5% vs. 15.6%, *p* < 0.001), and residual cytogenetic abnormalities before HCT (51.2% vs. 11.4%, *p* < 0.001). The distribution of donor type also significantly differed: more MRD^pos^ patients received matched unrelated donor transplants compared to MRD^neg^ patients (71.1% vs. 58.8%, *p* = 0.049). As a reflection of this, the type of GVHD prophylaxis varied between MRD^pos^ and MRD^neg^ patients (*p* = 0.046). There were also differences in maintenance therapy between groups (*p* < 0.001). In contrast, no significant differences were noted between the two groups with respect to patient sex, HCT-CI, ECOG performance status, blood count recovery before HCT, or year of transplantation (before vs. after 2017), donor age, donor CMV status, or graft cryopreservation status (fresh vs. cryopreserved; Table 1).

Across the entire cohort, the median follow-up time amongst survivors was 97 months (range, 12–225 months). 3-year relapse incidence, RFS, and OS was 29% (95% confidence interval: 25–34%), 61% (56–66%), and 67% (62–72%). 100-day and 18-month NRM was 3% (1–5%) and 8% (5–10%), respectively. Consistent with our previous studies, patients with pre-HCT MRD had higher risks of relapse and lower survival expectations at 3 years compared to MRD^neg^ patients (relapse: 63% [51–73%] vs. 21% [16–25%)], *p* < 0.001; RFS: 29% [20–41%] vs. 69% [64–74%], *p* < 0.001; OS: 40% [30–53%] vs. 73% [68–78%], *p* < 0.001) [36]. In contrast, there was no difference in 100-day (1% [0–6%] vs. 3% [2–5%], *p* = 0.4) or 18-month (7% [2–14%] vs. 8% [5–11%], *p* = 0.4) NRM between MRD^pos^ and MRD^neg^ patients. In our cohort, the cumulative incidence for acute GVHD grade 3–4 by day 100 was 14% (10–18%), whereas the 180-day cumulative incidence of moderate to severe chronic GVHD was 10% (7–13%). As observed in previous findings from our institution, there was no difference in relapse, RFS, OS, or NRM for patients receiving cryopreserved vs. fresh grafts [44].

### 3.2. Association Between CD34+ Cell Dose and Post-HCT Outcomes

MFC assessment of infused composition was available for almost 95% of all patients. For those with available data, the median dose of infused cellular product, including CD34+, T cells (CD3+, CD4+, and CD8+), Treg, B cells, NK cells and monocytes, is summarized in Table 2. CD34+ and CD3+ cell doses showed a weak positive correlation (r = 0.24, *p* < 0.001). There were no statistically significant differences in cellular graft doses between MRD^pos^ and MRD^neg^ patients. There was a weak association between increasing donor age and CD34+ dose (r = −0.11, *p* = 0.04), but not CD3+ dose (r = 0.05, *p* = 0.37). CD3+ doses were higher in fresh compared to cryopreserved grafts (266.3 [4.9–955.1] × 10^6^/kg vs. 215.7 [112.7–529.5] × 10^6^/kg, *p* = 0.031), while there was no statistically significant difference in CD34+ doses (*p* = 0.34).

We first analyzed the effect of doses of infused CD34+ cells on relapse, RFS, OS, and NRM, with graft cell doses used as a continuous variable. Among the 358 patients with available data on infused CD34+ doses, the median dose was 7.9 × 10^6^/kg (range: 1.0–38.6 × 10^6^/kg). Across the entire study cohort, the CD34+ cell dose did not show a statistically significant linear relationship between relapse risk (hazard ratio [HR, per 1.0 × 10^6^/kg increase in CD34+ cell dose] = 0.96 [0.91–1.01], *p* = 0.14), RFS (HR = 0.97 [0.93–1.01], *p* = 0.08), OS (HR = 0.98 [0.94–1.02], *p* = 0.28), and NRM (HR = 1.00 [0.95–1.06], *p* = 0.93).

To assess potential non-linear vs. linear relationships between CD34+ cell dose and post-HCT outcomes, we conducted restricted cubic spline analyses (Figure 1). These analyses suggested a non-linear relationship between infused CD34+ cell dose and relapse, RFS, and OS both in the entire cohort and when stratified by MRD status. Specifically, as depicted in Figure 1**,** in the entire cohort, there appeared to be a progressive decrease in relapse risk and improvement in RFS and OS with increasing CD34+ doses up to certain thresholds, above which the relapse risk no longer decreased. Using a maximally selected rank statistic, we identified a CD34+ cell dose threshold of 5.61 × 10^6^/kg for relapse risk and RFS, and 5.47 × 10^6^/kg for OS in the entire cohort. Recognizing that relapse is the key reason for failure of allogeneic HCT, we thus divided patients into two groups of “low” (≤5.61 × 10^6^/kg; *n* = 81 [23%]) and “high” (>5.61 × 10^6^/kg; *n* = 278 [77%]) CD34+ cell doses based on the maximally selected rank statistic for relapse. Interestingly, we observed a relationship between patients who received a “low” CD34+ cell dose and higher patient body weight, worse ECOG performance status, and slightly higher TRM scores (body weight: 83.7 vs. 78.0 kg, *p* = 0.025, ECOG 1–2: 80% vs. 63%, *p* = 0.0020; TRM score: 1.6 vs. 1.1, *p* = 0.0093) (Appendix A). Further studies are needed to confirm this finding and to examine this relationship and possible etiologies in greater depth. Otherwise, there were no significant differences between the high vs. low CD34+ groups with respect to patient-, disease-, or transplant- characteristics. As shown in Figure 2, patients who received low CD34+ doses had worse outcomes compared to those who received high CD34+ doses, with higher cumulative incidence of relapse at 3 years (41% [30–52%] vs. 26% [21–31%], *p* = 0.002) and lower 3-year RFS (44% [35–57%] vs. 65% [60–71%], *p* < 0.001) and 3-year OS (52% [42–64%] vs. 70% [65–677%], *p* = 0.001). After multivariable adjustment (including TRM score, adverse cytogenetic risk, and pre-HCT MRD status), a low CD34+ dose was associated with worse post-HCT outcomes, namely increased risk of relapse (HR = 1.70 [1.13–2.56], *p* = 0.011) as well as lower OS (HR = 1.64 [1.15–2.33], *p* = 0.0067) and RFS (HR = 1.72 [1.23–2.40], *p* = 0.0014; Table 3). Conversely, there was no association between low CD34+ cell dose and NRM (HR = 1.15 [0.62–2.15], *p* = 0.65).

In contrast, CD34+ cell dose was not associated with the development of grade 2–4 acute GVHD (continuous variable: HR [per 1.0 × 10^6^/kg increase in CD34+ cell dose] = 1.02 [0.98–1.05], *p* = 0.33; low vs. high CD34+ group: HR = 0.96 [0.71–1.31], *p* = 0.81) or grade 3–4 acute GVHD (continuous variable: HR =1.00 [0.91–1.10], *p* = 0.99; low vs. high CD34+ group: HR = 1.57 [0.83–2.98], *p* = 0.17). Likewise, there was no significant association between CD34+ dose and moderate to severe chronic GVHD when CD34+ cell dose was used as a continuous variable (HR = 1.02 [0.99–1.06], *p* = 0.21). However, there was a trend toward reduced moderate to severe chronic GVHD in patients receiving low CD34+ cell count grafts when patients were dichotomized into low vs. high CD34+ cell counts (HR = 0.65 [0.41–1.05], *p* = 0.076).

Recognizing that this specific cd34+ threshold of 5.61 × 10^6^/kg may be impractical for clinical utility, we conducted sensitivity analysis by evaluating cutpoints at both cd34+ doses of 5 × 10^6^/kg and 6 × 10^6^/kg. indeed, cd34+ doses above these two cutpoints were consistently associated with decreased post-hct relapse risk (for cd34+ dose cutpoint of 5 × 10^6^/kg: hr = 1.80 [1.18–2.76], *p* = 0.007; for cd34+ dose cutpoint of 6 × 10^6^/kg: hr = 1.61 [1.10–2.34], *p* = 0.014).

### 3.3. Effect of Lower CD34+ Doses on Post-HCT Outcomes Is Independent of MRD Status

We next sought to evaluate the impact of graft CD34+ dose on post-HCT outcomes in the context of pre-HCT MRD status. Overall, the shape of the restricted cubic spline models of the relationship between CD34+ cell dose and post-HCT outcomes for MRD^neg^ and MRD^pos^ cohorts mirrored the entire cohort, with expected increased variability given smaller sample sizes (Figure 1). Interestingly, in the MRD^pos^ cohort, there was an apparent “U”-shaped relationship with worse outcomes (i.e., increased risk of relapse and RFS) not only with very low CD34 doses, but also with very high CD34 doses, acknowledging small subset sizes and large confidence intervals. This was further explored with the optimal equal-hazard ratio method to estimate two optimal cutpoints with approximately similar log hazard value of relapse in the MRD^pos^ cohort, which found a similar “low” CD34+ cutpoint of 5.10 × 10^6^/kg with the entire cohort. However, the “very high” cutpoint (CD34+ dose >19.53 × 10^6^/kg) defined a cohort of only 10 patients, which we considered too small for reliable analysis.

We then calculated the cumulative incidence of relapse, RFS, OS, and NRM, categorizing patients by pre-HCT MFC MRD status and low vs. high CD34+ dose (Figure 3)**.** The optimal cutpoints for low vs. high CD34+ dose, as defined by maximally selected rank statistics, were broadly consistent not only between the MRD^neg^ and MRD^pos^ cohorts but also relative to the entire cohort (relapse: for MRD^neg^ patients = 5.59 × 10^6^/kg, for MRD^pos^ patients = 5.08 × 10^6^/kg; RFS: for MRD^neg^ patients = 5.78 × 10^6^/kg, for MRD^pos^ patients = 5.08 × 10^6^/kg; OS: for MRD^neg^ patients = 5.78 × 10^6^/kg, for MRD^pos^ patients = 4.06 × 10^6^/kg). Allowing for some degree of variability given small cohort sizes, the overall similarity in cutpoints suggested no substantial difference in optimal CD34+ thresholds between MRD^pos^ patients, MRD^neg^ patients and the entire cohort. Consequently, we adopted a uniform threshold of 5.61 × 10^6^/kg for subsequent analyses. When examining the MRD^pos^ subset of patients, we found that compared to patients given high CD34+ doses (MRD^pos^/high CD34+; *n* = 52), those who received low CD34+ doses (MRD^pos^/low CD34+; *n* = 20) trended toward, but did not reach statistical significance for, a higher risk of relapse (80% [52–93%] vs. 58% [43–70%], *p* = 0.073). This was likely due to limited statistical power from small cohort sizes. Similarly, there was no statistical difference in RFS or OS based on CD34+ doses among MRD^pos^ recipients (3-year RFS: for low CD34+ dose= 15% [5–43%], for high CD34+ dose= 34% [23–50%], *p* = 0.17; 3-year OS: for low CD34+ dose= 30% [15–59%], for high CD34+ dose= 45% [33–61%], *p* = 0.49). There was only one event for NRM among MRD^pos^ recipients of low CD34+ doses in our study.

In the MRD^neg^ subset of patients, low CD34+ doses (MRD^neg^/low CD34+; *n* = 61) were associated with increased relapse incidence and worse RFS and OS at 3-years compared to high CD34+ doses (for relapse: 29% [18–41%] vs. 18% [14–24%], *p* = 0.03; for RFS: 54% [43–69%] vs. 72% [66–78%], *p* = 0.002; for OS: 59% [48–73%] vs. 76% [71–82%], *p* = 0.002). However, there was no statistical difference in 100-day NRM (5% [1–12%] vs. 2% [1–5%], *p* = 0.20). In an interaction model, there was no significant interaction between pre-HCT MRD status and CD34+ dose groups for relapse (*p* = 0.87) or RFS (*p* = 0.62), suggesting that the effect of CD34+ dose groups on relapse rates and RFS did not depend on MRD status.

Like the findings from the entire cohort, CD34+ cell dose (both as a continuous variable and dichotomized as low vs. high group) was not associated with grade 3–4 acute GVHD or moderate to severe chronic GVHD in either the MRD^pos^ or MRD^neg^ subset of patients.

### 3.4. Association of Higher CD3+ Cell Doses with Increased Risk of Chronic but Not Acute GVHD

We next analyzed the effect of doses of infused CD3+ cells on relapse, RFS, OS, NRM, acute GVHD, and chronic GVHD. Among the 358 patients with available information on infused CD3+ doses, the median cell dose was 261.5 × 10^6^/kg (range: 4.9–955.1 ×10^6^/kg). In our cohort, there was no association between CD3+ cell dose as a continuous variable and relapse risk, RFS, OS, or NRM (relapse: HR [per 1.0 × 10^8^/kg increase in CD3+ cell dose] = 1.01 [0.89–1.16], *p* = 0.85); RFS: HR = 1.02 [0.92–1.14], *p* = 0.67; OS: HR = 0.98 [0.87–1.10], *p* = 0.73); NRM: HR = 1.05 [0.88–1.26], *p* = 0.57). In addition, restricted cubic spline analyses did not suggest a non-linear relationship between CD3+ dose and these post-HCT outcomes in the entire cohort or the MRD^neg^ cohort (Appendix A). Interestingly, there appeared to be a signal for a non-linear relationship between receiving low CD3+ doses (cutpoint = 286.6 × 10^6^/kg; for low CD3+ group, *n* = 45; for high CD3+ group, *n* = 27) and worse outcomes of relapse and RFS in the MRD^pos^ cohort. This association of low CD3+ dose with worse relapse, but not RFS, remained significant after multivariate adjustment (for relapse: HR= 0.36 [0.19–0.72], *p* = 0.004; for RFS: HR = 0.39 [0.19–0.72]. Given small cohort sizes, further studies will be needed prior to definitive conclusions.

CD3+ doses were not associated with grade 3–4 acute GVHD in univariate analysis (entire cohort: HR = 1.00 [0.81–1.25], *p* = 0.97; MRD^pos^ patients: HR = 1.08 [0.61–1.92], *p* = 0.79; MRD^neg^ patients: HR = 1.00 [0.79–1.27] *p* = 0.99). On the other hand, when analyzed as a continuous variable, CD3+ (HR = 1.15 [1.04–1.28], *p* = 0.009) doses were associated with risk of moderate–severe chronic GVHD (Table 4). When accounting for use of PTCy as GVHD prophylaxis, the CD3+ dose remained significantly associated with increased risk of moderate–severe chronic GVHD in the entire cohort (HR = 1.16 [1.04–1.29], *p* = 0.009) and in the MRD^neg^ subset of patients (HR = 1.21 [1.09–1.35], *p* < 0.001). Acknowledging the limitations of a small cohort size; however, we did not find a statistically significant association between CD3+ doses and development of moderate–severe chronic GVHD in univariate analysis in the MRD^pos^ subset of patients (HR = 0.81 [0.53–1.24], *p* = 0.33).

## 4. Discussion

Numerous studies have investigated how components of donor cell products, particularly CD34+ cell doses, impact outcomes of allogeneic HCT for adults with AML or other hematologic malignancies. Thus far, results have been mixed, with some (but not all) analyses suggesting higher graft CD34+ doses are associated with a decreased relapse incidence and, consequently, improved survival [8,9,10,17,19,20,22,23,24,25,26,27]. In our study, which we focused entirely on a homogeneous patient cohort of adults with AML receiving HLA-matched related or unrelated donor allografts after MAC, we found evidence of a statistically significant, and likely clinically meaningful, association between higher CD34+ cell dose and improved relapse risk, RFS, and OS. Using cubic spline analyses and maximally selected rank statistics, our results indicated that this association was best described using a non-linear relationship: CD34+ doses up to 5–6 × 10^6^/kg correlated with progressively lower risk of relapse, but beyond this threshold, no longer decreased. This relationship appeared to be independent of pre-HCT MRD status. In addition, we found that CD3+ T cell doses in our cohort were associated with higher rates of moderate to severe chronic GVHD, but not acute GVHD. Together, our data support that the cellular composition of the donor graft impacts outcomes in adults with AML undergoing allogeneic HCT in first morphologic remission after MAC.

In the previous retrospective analysis by the Blood and Marrow Transplant Clinical Trials Network (BMT CTN), the use of CD34-selected T cell-depleted grafts (with target graft composition of CD34+ cells > 5 × 10^6^/kg and CD3+ cells < 1 × 10^5^/kg) did not result in reduced relapse rates or improved RFS or OS at 2-years, but did lead to a significantly lower incidence of chronic GVHD in 44 patients when compared to an overall similar cohort of 84 patients who received T-replete PBSC allografts and pharmacologic immune suppression therapy for GVHD prophylaxis [45]. In contrast, other retrospective studies have shown a positive association between CD34+ cell dose in T-replete allografts and reduced risk of relapse, with cutoffs ranging from 4 to 6 × 10^6^/kg [10,17,19,46].

In line with these latter studies, we found that CD34+ cell doses > 5.61 × 10^6^/kg resulted in decreased relapse risk and improved RFS and OS, without an effect on NRM. Further sensitivity analysis was applied, which showed that this benefit was maintained with CD34+ doses above the range of 5–6 × 10^6^/kg. In addition, the benefits of higher CD34+ cell doses were independent of MRD status. In subset analysis of MRD^pos^ and MRD^neg^ patients, recipients of higher CD34+ cell doses had similar patterns in decreased relapse risk and improved RFS, respectively, although notably our analyses were limited by small cohort sizes. Interestingly, cubic spline modeling of the MRD^pos^ cohort suggested an intermediate CD34+ dose that optimized post-HCT outcomes; however, given limited cohort sizes at extremely high doses, this was considered too small for reliable analysis and interpretation. Other studies have suggested similar findings of worse post-HCT outcomes with higher CD34+ doses, although the exact cutoff described is varied, anywhere from 3 × 10^6^/kg to 8 × 10^6^/kg [17,22,23]. Currently, more studies are needed prior to definitive conclusion on the risks and/or benefits of higher CD34+ cell doses.

Similarly to previous studies, our analyses also revealed that higher CD3+ T cell doses were associated with increased risk of moderate to severe chronic GVHD, without a significant impact on relapse risk, RFS, OS, NRM or acute GVHD [27,47]. This relationship even persisted after taking the use of PTCy as GVHD prophylaxis into consideration. Unlike other studies, we did not find that higher CD34+ doses resulted in a statistically significant increased risk of chronic GVHD in our analyses, although there was a trend toward significance [17,19,20,23]. However, given the association between CD34+ and CD3+ cell doses in the graft, this trend may reflect the influence of CD3+ T cells. In addition, our data suggested a possible association between CD3+ doses > 286 × 10^6^/kg and reduced relapse risk in the MRD^pos^ cohort. However, the small sample size limits the strength of this observation, and additional studies are needed to validate this finding.

To our knowledge, this is the first study that evaluated the impact of graft cellular composition on post-HCT outcomes in the context of pre-HCT MRD status. Our study is comprehensive, including essentially all adults undergoing myeloablative allogeneic HCT in first remission with 10/10 HLA-matched related or unrelated peripheral blood stem cells at our institution as all patients are asked during their pre-HCT workup—with almost all providing informed consent—for the use of their clinical data in retrospective research like the one described herein. In addition, MFC MRD testing is routinely performed on pre-HCT bone marrow specimens and detailed flow cytometric information on the cellular composition of donor cells was available for almost all patients, although missing data was more common among patients transplanted in earlier years. On the other hand, several limitations of our study must be acknowledged. Most importantly, this was a retrospective analysis of patients assigned to different transplant protocols in a (largely) non-randomized fashion at a single institution, and our findings may not easily be extrapolated. As such, demonstrating a causal relationship between graft composition and post-HCT outcomes outside of a controlled clinical trial is challenging, and most treatment assignments were in a non-randomized fashion. Second, because mutational profiles were only available for a subset of patients, our ability to account for ELN 2022 disease risk was limited. And fourth, there was no uniform approach to post-HCT leukemia recurrence; approaches included expedited withdrawal of immunosuppressive agents, donor lymphocyte infusions, treatment with azanucleosides or molecularly targeted agents, administration of intensive chemotherapy, or combinations thereof.

Acknowledging these limitations, our data identify a non-linear relationship between CD34+ cell dose and relapse risk among AML patients undergoing myeloablative allogeneic HCT with HLA-matched related or unrelated peripheral blood stem cells. This relationship was observed among both patients without and those with pre-HCT MFC MRD. Our findings suggest that, for such patients, a minimum CD34+ cell dose of 5–6 × 10^6^/kg should be targeted for optimal post-HCT outcomes. We do not suggest a “cap” on higher CD34+ cell doses in MRD^pos^ cohort based on the limited sample size of our cohort at higher doses, with future larger studies needed prior to definitive recommendations. Since higher CD3+ cell doses were independently associated with the risk of moderate to severe chronic GVHD, and there was a weak association between CD34+ and CD3+ cell counts, cell counts other than CD34+ should be considered as well for optimal treatment outcomes. Additional studies will be necessary to test whether similar principles regarding CD34+ and CD3+ cell doses apply for other donor cell scenarios, and what other graft cell components might impact outcomes in AML patients receiving HLA-matched donor cells.

## 5. Conclusions

In sum, our study demonstrates that the CD34+ and CD3+ cellular composition of the donor graft significantly influences post-HCT outcomes in adults with AML undergoing myeloablative HCT with HLA-matched related or unrelated peripheral blood stem cells. In particular, the benefit of receiving CD34+ cell doses above the range of 5–6 × 10^6^/kg was independent of pre-HCT MRD status, supporting the use of a minimum threshold for CD34+ cells for most patients. Additionally, our findings reinforce prior evidence that higher CD3+ cell doses are associated with increased risk of moderate to severe chronic GVHD, independent of relapse risk or survival outcomes. CD34+ and CD3+ cell doses should be considered in graft selection for optimal post-HCT outcomes. Importantly, future prospective observational studies that systematically capture graft CD34+ and CD3+ doses are needed to further clarify the impact of graft cell composition on post-HCT outcomes for this patient population.

## Figures and Tables

**Figure 1 cancers-17-02323-f001:**
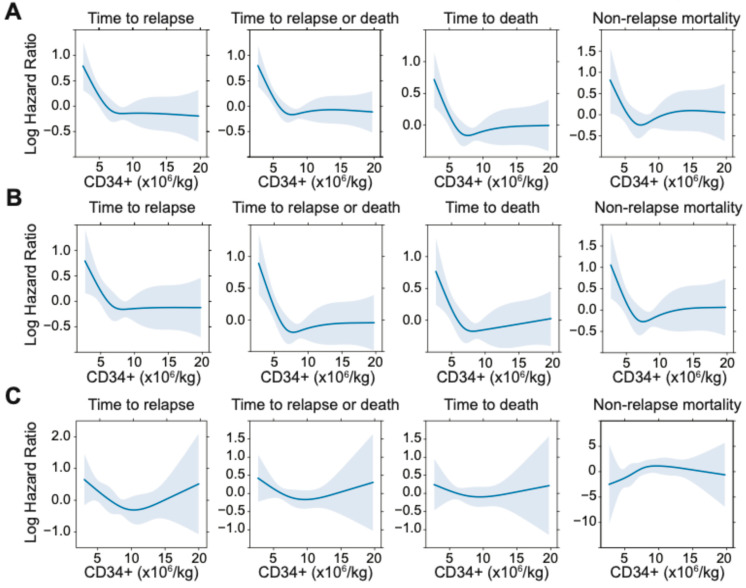
Restricted cubic spline functions for post-HCT outcomes of relapse, relapse-free-survival, overall survival, and non-relapse mortality in (**A**) the entire cohort, (**B**) the subset of 308 patients without pre-HCT MFC MRD, and (**C**) the subset of 76 patients with pre-HCT MFC MRD.

**Figure 2 cancers-17-02323-f002:**
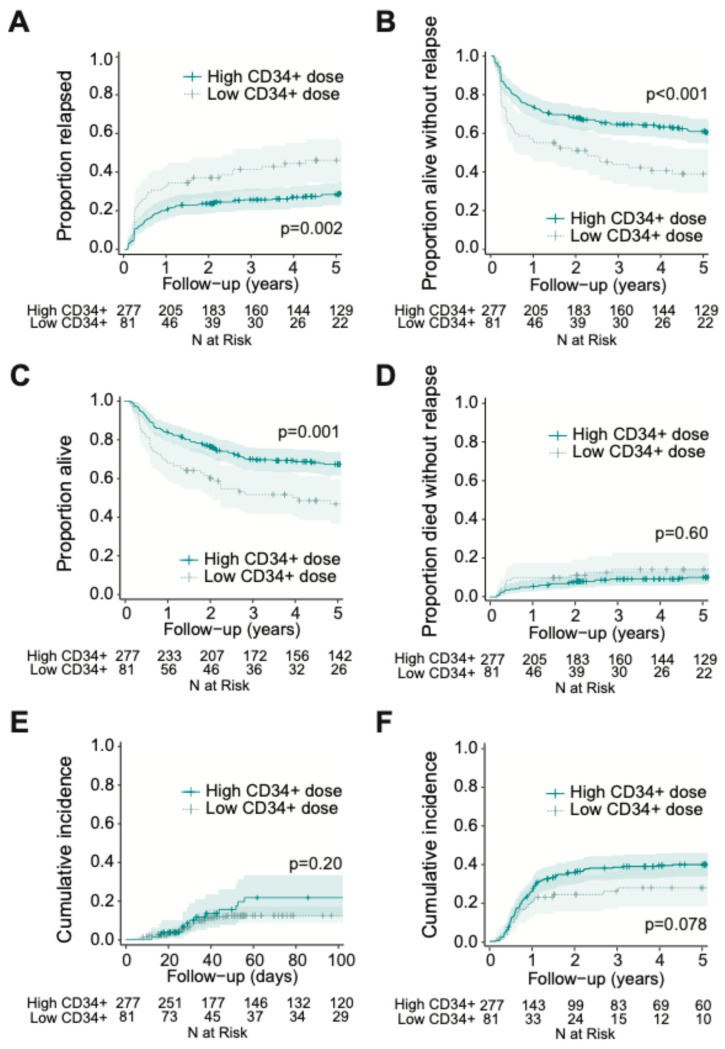
Post-HCT outcomes, including (**A**) risk of relapse, (**B**) RFS, (**C**) OS, (**D**) risk of NRM, (**E**) grade 3 to 4 acute GVHD, and (**F**) moderate to severe chronic GVHD, for adults with AML undergoing allogeneic HCT while in first morphologic remission, stratified by CD34+ cell graft dose (threshold = 5.61 × 10^6^/kg).

**Figure 3 cancers-17-02323-f003:**
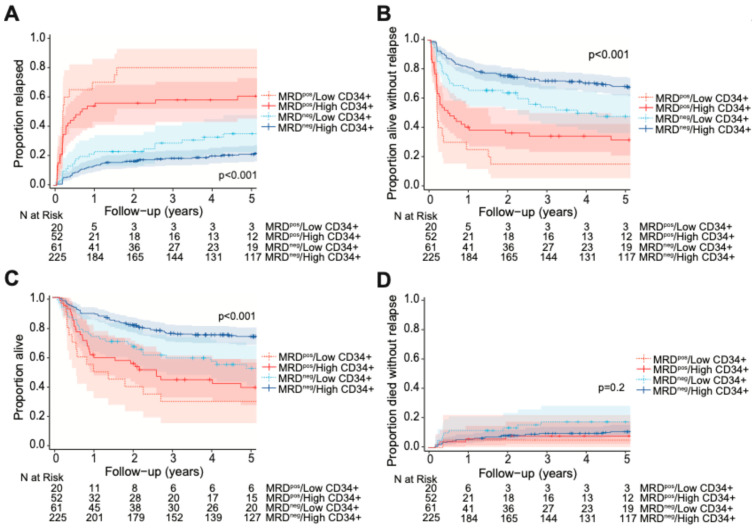
Post-HCT outcomes, including (**A**) risk of relapse, (**B**) RFS, (**C**) OS, and (**D**) risk of NRM, for adults with AML undergoing allogeneic HCT while in first morphologic remission, stratified by CD34+ cell graft dose and pre-HCT MFC MRD status.

**Table 1 cancers-17-02323-t001:** Patient, disease, and transplantation characteristics of the study population, stratified by MRD status.

	All Patients (*n* = 384)	MRD^pos^ Patients (*n* = 76)	MRD^neg^ Patients (*n* = 308)	*p*-Value
**Patient characteristics**				
**Age, median (range)**	51 (18–74)	54 (18–72)	50 (19–74)	0.005
**Body weight, in kg (range)**	79.7 (47.3–187.5)	79.0 (47.8–135.0)	79.8 (47.3–187.5)	0.70
**Male sex, *n* (%)**	194 (50.5%)	41 (53.9%)	153 (49.7%)	0.50
**HCT-CI, *n* (%)**				0.70
Low (0–1)	175 (45.6%)	37 (48.7%)	138 (44.8%)	
Intermediate (2–3)	125 (32.6%)	25 (32.9%)	100 (32.5%)	
High (≥4)	77 (20.1%)	12 (15.8%)	65 (21.1%)	
Unknown	7 (1.8%)	2 (2.6%)	5 (1.6%)	
**ECOG PS, *n* (%)**				0.55
0	125 (32.5%)	27 (35.5%)	98 (31.8%)	
1–2	258 (67.2%)	49 (64.5%)	209 (67.9%)	
Unknown	1 (0.3%)	0 (0%)	1 (0.3%)	
**TRM score, median (range)**	1.2 (0.04–17.4)	2.1 (0.04–17.4)	1.1 (0.04–12.2)	<0.001
**Disease characteristics**				
**Disease type**				<0.001
AML	316 (82.3%)	50 (65.8%)	266 (86.4%)	
MDS/AML	68 (17.7%)	26 (34.2%)	42 (13.6%)	
**Cytogenetic risk, *n* (%)**				<0.001
Favorable	12 (3.1%)	0 (0%)	12 (3.9%)	
Intermediate	245 (63.8%)	36 (47.40%)	209 (67.9%)	
Adverse	115 (29.9%)	38 (50.0%)	77 (25.0%)	
Unknown	12 (3.1%)	2 (2.6%)	10 (3.2%)	
**Secondary AML**	75 (19.5%)	27 (35.5%)	48 (15.6%)	<0.001
**Blood counts before HCT, *n* (%)**				0.91
CR	292 (76.0%)	57 (75.0%)	235 (76.3%)	
CRh/CRi	80 (20.8%)	17 (22.4%)	63 (20.5%)	
MLFS	12 (3.1%)	2 (2.6%)	10 (3.2%)	
**Karyotype before HCT, *n* (%)**				<0.001
Normalized	147 (38.3%)	13 (15.6%)	134 (43.5%)	
Abnormal	76 (19.8%)	42 (51.2%)	35 (11.4%)	
Non-informative *	161 (41.9%)	27 (32.9%)	139 (45.1%)	
**Transplantation characteristics**				
**Transplant year, *n* (%)**				0.12
2006–2017	254 (66.1%)	56 (73.7%)	198 (64.3%)	
2017–2023	130 (33.9%)	20 (26.3%)	110 (35.7%)	
**Donor age, median (range)**	35 (17–70)	36 (18–69)	35 (17–70)	0.24
**Donor type, *n* (%)**				0.049
HLA-identical sibling donor	149 (38.8%)	22 (28.9%)	127 (41.2%)	
10/10 matched unrelated donor	235 (61.2%)	54 (71.1%)	181 (58.8%)	
**Donor CMV status**				0.56
Positive	165 (43.0%)	28 (36.8%)	137 (44.5%)	
Negative	217 (56.5%)	48 (63.2%)	169 (54.9%)	
Unknown	2 (0.6%)	0 (0%)	2 (0.6%)	
**Graft status**				0.80
Fresh	322 (83.9%)	63 (82.9%)	259 (84.1%)	
Cryopreserved	62 (16.1%)	13 (17.1%)	49 (15.9%)	
**GVHD prophylaxis, *n* (%)**				0.046
CNI + MMF ± Sirolimus	64 (16.7%)	20 (26.3%)	44 (14.3%)	
CNI + MTX ± Other(s)	272 (70.8%)	46 (60.5%)	226 (73.4%)	
PTCy	37 (9.6%)	9 (11.8%)	28 (9.1%)	
Others	11 (2.9%)	1 (1.3%)	10 (3.2%)	
**Maintenance therapy, *n* (%)**				<0.001
None	276 (71.9%)	40 (52.6%)	236 (76.6%)	
FLT3 inhibitor	24 (6.3%)	0 (0%)	24 (7.98%)	
HMA ± VEN	5 (1.3%)	3 (3.9%)	2 (0.6%)	
HMA + FLT3 inhibitor	2 (0.5%)	2 (2.6%)	0 (0%)	
Unknown	77 (20%)	31 (40.8%)	46 (14.9%)	

* Normal cytogenetics in patient with cytogenetically normal AML or missing cytogenetics at diagnosis. Abbreviations: CMV, cytomegalovirus; CNI, calcineurin inhibitor; CR, complete remission; CRh, CR with partial hematologic recovery; CRi, CR with incomplete hematologic recovery; ECOG PS, Eastern Cooperative Oncology Group performance status; FLT3, fms-like tyrosine kinase 3; GVHD, graft-versus-host disease; HCT, hematopoietic cell transplantation; HLA, human leukocyte antigen; HMA, hypomethylating agent; MDS, myelodyplastic syndrome; MLFS, morphologic leukemia-free state; MMF, mycophenolate mofetil; MRD^pos^, measurable residual disease positive; MRD^neg^, measurable residual disease negative; MTX, methotrexate; PTCy, post-transplantation cyclophosphamide; TRM, treatment-related mortality; VEN, venetoclax.

**Table 2 cancers-17-02323-t002:** Infused graft characteristics of the study population, stratified by MRD status.

	All Patients (*n* = 384)	MRD^pos^ Patients (*n* = 76)	MRD^neg^ Patients (*n* = 308)	*p*-Value
CD34+ cell dose, ×10^6^/kg, median (range), *n* = 358	7.9 (1.0–38.6)	7.7 (2.1–20.9)	8.0 (1.0–38.6)	0.45
CD3+ T cell dose, ×10^6^/kg, median (range), *n* = 358	261.5 (4.9–955.1)	252.3 (32.5–573.8)	262.7 (4.9–955.1)	0.50
CD4+ T cell dose, ×10^6^/kg, median (range), *n* = 357	164.2 (0.08–678.0)	158.4 (7.5–415.4)	166.9 (0.08–678.0)	0.97
CD8+ T cell dose, ×10^6^/kg, median (range), *n* = 357	91.1 (0.5–299.3)	81.9 (2.7–264.5)	92.9 (0.5–299.3)	0.34
Treg cell dose, ×10^5^/kg, median (range), *n* = 298	6.1 (0.05–56.4)	8.0 (0.05–53.6)	6.0 (0.05–56.4)	0.67
B cell dose, ×10^6^/kg, median (range), *n* = 341	61.1 (0.003–289.9)	57.4 (7.9–186.8)	62.4 (0.003–289.9)	0.49
NK cell dose, ×10^6^/kg, median (range), *n* = 341	25.5 (0.01–108.5)	28.1 (0.07–104.3)	25.0 (0.01–108.5)	0.57
Monocyte cell dose, ×10^6^/kg, median (range), *n* = 352	228.9 (5.7–577.9)	228.4 (61.9–564.9)	230.3 (5.7–577.9)	0.91

Abbreviations: NK cell, natural killer cell; Treg, regulatory T cells.

**Table 3 cancers-17-02323-t003:** Multivariate regression models of clinical outcomes (entire study cohort).

	Relapse Risk		NRM		Risk of Relapse/Death		Risk of Death	
	HR (95% CI)	*p*-Value	HR (95% CI)	*p*-Value	HR (95% CI)	*p*-Value	HR (95% CI)	*p*-Value
Patient weight (by 10 kg change)	Not included in model		1.12 (1.00–1.26)	0.059	Not included in model		Not included in model	
TRM score (by 5 point change)	1.14 (0.73–1.78)	0.56	1.41 (0.84–2.37)	0.19	1.36 (0.99–1.89)	**0.059**	1.58 (1.14–2.19)	**0.0059**
Adverse cytogenetic risk (ref: intermediate)	1.72 (1.06–2.88)	**0.030**	0.51 (0.26–0.97)	**0.039**	1.14 (0.77–1.71)	0.50	0.99 (0.65–1.51)	0.96
Pre-HCT MRD^pos^ (ref: MRD^neg^)	3.14 (1.96–5.02)	**<0.001**	Not included in model		2.52 (1.72–3.70)	**<0.001**	1.76 (1.17–2.65)	**0.0064**
Pre-HCT karyotype (ref: normalized) Not normalized	1.51 (0.85–2.68)	0.16	Not included in model		1.31 (0.84–2.06)	0.22	1.40 (0.88–2.24)	0.16
MLFS before HCT (ref: CR)	Not included in model		2.76 (0.77–9.89)	0.12	Not included in model		Not included in model	
Year of transplantation—before 2017 (ref: after 2017)	Not included in model		Not included in model		Not included in model		1.39 (0.93–2.06)	0.11
Low (≤5.61 × 10^6^) CD34+ dose group (ref: >5.61 × 10^6^)	1.70 (1.13–2.56)	**0.011**	Not included in model		1.72 (1.23–2.40)	**0.0014**	1.64 (1.15–2.33)	**0.0067**

*P*-values in bold are statistically significant (*p* < 0.05). Abbreviations: CR, complete remission; HCT, hematopoietic cell transplantation; MLFS, morphologic leukemia-free state; MRD^pos^, measurable residual disease positive; MRD^neg^, measurable residual disease negative; NRM, non-relapse mortality; TRM, treatment-related mortality.

**Table 4 cancers-17-02323-t004:** Univariate and multivariate regression models of moderate to severe chronic GVHD.

	Univariate		Multivariate	
	HR (95% CI)	*p*-Value	HR (95% CI)	*p*-Value
Age (by decade)	1.12 (0.98–1.28)	0.094	Not included in model	
ECOG performance status	0.82 (0.57–1.17)	0.27	Not included in model	
HCT before 2017 (ref: after 2017)	0.82 (0.58–1.16)	0.26	Not included in model	
10/10 HLA-matched unrelated donor (ref: sibling donor)	1.22 (0.86–1.75)	0.27	Not included in model	
Donor CMV positive (ref: donor CMV neg)	0.77 (0.54–1.10)	0.15	Not included in model	
PTCy (ref: CNI + MMF ± Sirolimus)	0.23 (0.07–0.78)	**0.019**	0.17 (0.04–0.71)	**0.015**
CD3+ T cell dose, (per 1.0 × 10^8^/kg increase)	1.15 (1.04–1.28)	**0.009**	1.16 (1.04–1.29)	**0.009**

*p*-values in bold are statistically significant (*p* < 0.05). Abbreviations: CMV, cytomegalovirus; CNI, calcineurin inhibitor; ECOG, Eastern Cooperative Oncology Group, GVHD, graft-versus-host disease; HCT, hematopoietic cell transplantation; HLA, human leukocyte antigen; MMF, mycophenolate mofetil; PTCy, post-transplantation cyclophosphamide.

## Data Availability

For original, de-identified data, please contact the corresponding author (rwalter@fredhutch.org).

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
