# Peer review of "CD34+ Cell Dose, Measurable Residual Disease, and Outcome After Myeloablative HLA-Matched Peripheral Blood Hematopoietic Cell Transplantation for Adults with Acute Myeloid Leukemia"

_cancers, 2025, doi:10.3390/cancers17142323_

Round 1
Reviewer 1 Report
Comments and Suggestions for Authors
The manuscript is well written and can be improved with consideration for the following.
- Please clarify if the pts designated as MDS/AML had secondary AML with preceding diagnosis of MDS. (Line 86-87)
- Since the manuscript focuses on myeloablative conditioning, please remove reference to non-MAC regimens in line 126.
- In the statistical section there is mention of "missing" category for pts without CD 34+ve cell dose. There is no mention of how many pts had the CD 34+ve cell count data missing, consider excluding those pts from analysis if there are patients with missing information.
- Since a small number of pts received post-transplant maintenance which could be a confounder for relapse, OS etc., Consider excluding those pts for analysis.
- There were 12 pts with favorable risk who were MRD negative but still underwent transplant... what the reason for the pts to undergo transplant in that situation? Table 1
- Table 3 Decimal point is missing for PRE-HCT karyotype column 1
Reviewer 2 Report
Comments and Suggestions for Authors
This manuscript looks at 385 AML pts transplanted with MAC PBSC grafts at a single centre over 17 yrs, and ties CD34⁺ dose (and to a lesser extent CD3⁺) to relapse, survival and GVHD, while in a sub group analysis determines if cell dose impacts pre‑transplant MRD. They should be commended for analysing by the non‑linear “threshold”: above roughly 5‑6x10⁶ CD34⁺/kg the relapse curve flattens, and this holds whether there is MRD or not. CD3⁺ dose, by contrast, increases chronic GVHD. In terms of the cohort size, it is reasonably sized, with a homogenous conditioning platform, MRD uniformly assessed, and with long follow-up. The statistical modelling was well planned.
That said, several other issues bear further consideration. Being single‑centre and including patients transplanted between 2006‑2023 means significant changes in supportive care and MRD technology may be a cause of bias; the year of HCT only partly addresses this is and there is no test of proportional hazards over treatment eras.
Incomplete mutational data and ELN-2022 classification. ELN risk requires multiple genes; mutational testing was available only for a subset. The authors should address this by specifying whether the cases of “missing” cytogenetic information correspond to the mutational data which likely was not available in 2006, or where missing was due to failure of cytogenetics etc.
How many of CD34⁺/CD3⁺ and several baseline variables had missing data? The authors should indicate the number with missing data.
Low CD34⁺ dose pts also carried higher TRM score and ECOG, suggesting selection rather than pure biological effect, and this needs examination, maybe with inverse probability of treatment weighting/IPTW or a sensitivity analysis. Alternatively the authors may wish to address this in the limitations section.
Minor issues:
- Figure legends: Abbreviations such as TRM, CRi are not defined in corresponding legends.
- Figure 2B: Threshold (5.61x10⁶/kg) not stated in the figure/legend.
In sum, the work is relevant, original and could influence graft-collection targets, but has some methodological issues. I would suggest that the issues being raised in this review be addressed before publication.
Reviewer 3 Report
Comments and Suggestions for Authors
This is a well-written manuscript on how the graft composition impact patient outcomes after transplantation for AML. I have the following observations:
- My main concern is that demonstrating a causal relationship between graft composition and outcomes is challenging outside of controlled clinical trials, As the authors correctly stated in the discussion, this analysis cannot account for events after transplantation, such as the timing of withdrawal immunosuppressive agents, the infusion of DLI and maintenance treatment. I wonder if it would be possible to perform a sub-analysis focusing on a group of patients selected who have homogeneous post-transplant treatment
- I believe that the outcome graft relapse-free survival (GRFS) should be evaluated considering the aims of this study. GRFS is likely the most critical outcome for assessing the effect of graft composition in terms of CD34+ and CD3+ cells on post-transplant results.
- It is unclear how the findings of this study may help guide decisions about donor cell selection (see simple summary, lines 28 and 29). Please provide clarification on this point.
- What types of studies do the authors propose should be conducted in order to address this significant question, particularly considering the involvement of healthy donors? The authors might share their perspective on how to approach this clinical issue in future studies.
Round 2
Reviewer 3 Report
Comments and Suggestions for Authors
The authors have addressed the concerns I raised in my previous review. Although the methodology remains a clear limitation in terms of the reliability of results, they have provided a more detailed comment on the main limitations (points 1 and 2). Overall, the revised version of the manuscript is improved.
Author Response
Comments 1: The authors have addressed the concerns I raised in my previous review. Although the methodology remains a clear limitation in terms of the reliability of results, they have provided a more detailed comment on the main limitations (points 1 and 2). Overall, the revised version of the manuscript is improved.
Response 1: We appreciate the thoughtful concerns raised by the reviewer and their time and efforts to improve our manuscript.